# Green Synthesis and Incorporation of Sericin Silver Nanoclusters into Electrospun Ultrafine Cellulose Acetate Fibers for Anti-Bacterial Applications

**DOI:** 10.3390/polym13091411

**Published:** 2021-04-27

**Authors:** Mujahid Mehdi, Huihui Qiu, Bing Dai, Raja Fahad Qureshi, Sadam Hussain, Muhammad Yousif, Peng Gao, Zeeshan Khatri

**Affiliations:** 1National Engineering Laboratory for Modern Silk, College of Textile and Clothing Engineering, Soochow University, Suzhou 215123, China; mujahid11te83@gmail.com (M.M.); kitty.huihui@aliyun.com (H.Q.); db@szdfhj.cn (B.D.); sadam11te75@gmail.com (S.H.); muhammadyousif.te72@gmail.com (M.Y.); 2Center of Excellence in Nanotechnology and Materials, Mehran University of Engineering and Technology, Jamshoro 76060, Pakistan; raja.ashraf@faculty.muet.edu.pk; 3Department of Textile Engineering, Mehran University of Engineering and Technology, Jamshoro 76060, Pakistan

**Keywords:** silver nanoclusters, sericin, cellulose acetate, electrospinning, Antibacterial Nanofibers

## Abstract

Fiber based antibacterial materials have gained an enormous attraction for the researchers in these days. In this study, a novel Sericin Encapsulated Silver Nanoclusters (sericin-AgNCs) were synthesized through single pot and green synthesis route. Subsequently these sericin-AgNCs were incorporated into ultrafine electrospun cellulose acetate (CA) fibers for assessing the antibacterial performance. The physicochemical properties of sericin-AgNCs/CA composite fibers were investigated by transmission electron microscopy (TEM), field emission electron microscopy (FE-SEM), Fourier transform infrared spectroscopy (FTIR) and wide X-ray diffraction (XRD). The antibacterial properties of sericin-AgNCs/CA composite fibers against *Escherichia coli* (*E. coli*) and *Staphylococcus aureus* (*S. aureus*) were systematically evaluated. The results showed that sericin-AgNCs incorporated in ultrafine CA fibers have played a vital role for antibacterial activity. An amount of 0.17 mg/mL sericin-AgNCs to CA fibers showed more than 90% results and elevated upto >99.9% with 1.7 mg/mL of sericin-AgNCs against *E. coli*. The study indicated that sericin-AgNCs/CA composite confirms an enhanced antibacterial efficiency, which could be used as a promising antibacterial product.

## 1. Introduction

Globally, a new and emerging variety of infectious diseases have brought the human health under a big threat. In most of the cases, the bacterial infection not only cause severe sickness accompanied by developing viral influenza (such as SARs and avian flu), but these microorganisms can also be spread into the environment making it rather unsafe [1,2]. For this reason, exploring new antimicrobial agents have gained much attention in past few decades. Generally, the metallic nanoparticles (Ag, Au, Cu, and Pt) and metallic oxides nanoparticles (Ag_2_O, CuO, ZnO, TiO_2_, Fe_2_O_3_, and SiO_2_) has been reported for antibacterial applications [3]. Amongst these nanoparticles, the silver nanoparticles are notably preferred due to their strong antimicrobial efficiency against microorganisms [4]. It is believed that the bacterial cell can be destroyed upon its the interaction with the Ag ion. The positively charged Ag ions, due to their strong affinity to sulfur proteins, adhere to the cell wall and cytoplasmic membrane [5,6,7]. This results into the blockage of bacterial respiratory system and demolishing the cell production. Subsequently this leads to the remarkable decline into the growth of further bacteria [8,9].

In a comparison with Ag particles, Ag ultra-small nanoparticles are improved in chemical stability, heat resistance, release of Ag ions, and durability, which will ultimately bring more opportunities for antimicrobial products [10]. Previous investigations revealed that the size [11], shape [12], surface coating [13], and surface charge [14] of silver ultra-small nanoparticles can significantly affect antibacterial efficacy. The novel metal (Au, Ag, Cu, and Pt) nanoclusters (NCs) composed of few to hundreds of atoms exhibited ultra-small sizes (<5 nm) that approach to the fermi wavelength of the conduction electrons. They show discrete energy levels, resulting in obvious fluorescence and molecule-like properties [15,16]. Thus, these NCs possess very unique properties than that of traditional metal nanoparticles and have attracted great attraction in recent years. These NCs have been extensively investigated in biolabeling and bioimaging, medicine, catalysis, and nanoelectronics [17,18]. Therefore, it is anticipated that silver nanoclusters (AgNCs) may also exhibit superior antimicrobial properties. Recently, Jian-Cheng group has reported the antibacterial performance of ultra-small DHLA-AgNCs and the results showed that silver ions of rich NCs could act as efficient nano reservoirs for antibacterial activity [19,20]. Sericin is extracted from silk fiber (*Bombyx mori* silkworm) and present in the outer layer of silk fiber. Sericin contains 18 types of amino acids, such as serine, glycine, lysine, etc. Sericin is a biocompatible and biodegradable natural biopolymer exhibiting antioxidant, moisture absorption, antibacterial and UV resistance properties [21,22].

Recently, safe and advanced non-woven nanofibrous dressings have received tremendous attention in antibacterial applications [23]. It is also stated that nanofibers have remarkable properties such as high interconnected porosity, large surface area to volume ratios [24,25]. To enhance the stability and therapeutic efficacy, Ag nanoparticles has been incorporated into various polymeric nanofibers including cellulose acetate (CA) [26], polyurethane [27], poly (ethylene oxide) [28], poly (vinyl alcohol) [29], and poly (lactic-co-glycolic acid) [30]. While CA contains polysaccharides groups from cellulose materials, it is a commonly used biopolymer and promising material for medical applications. Many conventional metal nanoparticles incorporating into electrospun nanofibers has been prepared. However, there is a considerable need to enhance antibacterial efficiency of electrospun membrane and its biocompatibility. Due to their ultra small size, it is assumed that metal nanoclusters can fulfil the above purpose.

In the present study, sericin mediated silver nanoclusters were prepared and incorporated into ultrafine CA nanofibers in order to assess the antibacterial activity of sericin-AgNCs/CA composite membrane against against *S. aureus* and *E. coli*. It was found that there is significant effect of sericin-AgNCs on the performance of antibacterial activity due to ultra-small size of Ag particles. The physicochemical properties of sericin-AgNCs incorporated into ultrafine CA fibers were examined with transmission electron microscopy (TEM), field emission electron microscopy (FE-SEM), Fourier transform infrared spectroscopy (FTIR) and wide X-ray diffraction (XRD). To the best of our knowledge, sericin-AgNCs were synthesized for the very first time, and incorporated into ultrafine CA fibers for antibacterial application. 

## 2. Experimental 

### 2.1. Materials

Cellulose acetate (CA, 30 KDa), Silver Nitrate (AgNO_3_, 99.9%), and sericin were purchased from Sigma Aldrich Co., Ltd. (St. Louis, MO, USA). Sodium hydroxide (NaOH, 99%), Acetone (C_3_H_6_O, 99.5%) and Dimethyl formamide (C_3_H_7_NO, 99.5%) were purchased from Sinopharm Chemical Reagent Co., Ltd. (Shanghai, China). Ultra-pure water was used during all experiments. 

### 2.2. Synthesis of Sericin-AuNCs

The template of Ag Nanoclusters with sericin was synthesized by using AgNO_3_ regent. For this, 20 mL of sericin (50 mg/mL) and 20 mL of AgNO_3_ (10 mM) was prepared separately and both solutions were transferred into an incubator at 37 °C for 10 min. Afterwards, both solutions were mixed together and shaken gently to get a homogenous solution of the mixture. Then, 2 mL NaOH (1 M) was added in the mixture and kept on stirring for 3 min. The solution was later incubated at 37 °C for 16 h and then placed further for 12 h into refrigerator at −20 °C. Finally, the sample was placed for freeze drying for 2 days to achieve Sericin-AgNCs in powder form.

### 2.3. Electrospinning 

The ultrafine CA fibers were fabricated according to our previous method [31]. Briefly, the polymer solution consists of 17% CA polymer was dissolved in acetone: DMF with ratio of 2:1 respectively. Then, different amounts of synthesized sericin-AgNCs powder (0.17 mg/mL, 0.85 mg/mL and 1.7 mg/mL) was added into the polymeric solution and stirred for 24 h in order to get homogeneous solution. The prepared solution was filled in a 5 mL syringe and placed vertically on micro-injection pump (LSP02-1B, Baoding longer precision pump Co., Ltd., Baoding, China). The feed rate for the solution was adjusted at a constant and controllable rate of 0.5 mL/h. High voltage of 10 kV was applied using power supply (DWP303-1AC, Tianjin Dongwen High Voltage Co., Baoding, China). The distance between needle and collector was set at 15 cm. The fibers were collected on aluminum foil and after collecting fibers, samples were dried in open air for 24 h. 

### 2.4. Antibacterial Activity

The antibacterial performance was analyzed using the common shake flask protocol [32]. This protocol is specifically designed for specimen’s treatment with non-releasing antibacterial agents under dynamic contact conditions. *Escherichia coli* (BUU25113) and *Staphylococcus aureus* (B-sub 168) were cultivated in nutrient broth at 37 °C for 18 h and the prepared solution was examined by broth dilution method. The number of viable bacterial cells reached 1 × 10^9^ cfu/mL. During four times serial dilution with 0.03 mol/L PBS, the number of viable bacterial cells is adjusted 3 × 10^5^ cfu/mL to 4 × 10^3^ cfu/mL. After obtaining the desired growth of bacterial cells, 0.05 g sample (CA fibers) was poured into conical flask containing 65 mL of 0.3 mM PBS solution and 5 mL of the prepared solution containing bacterial growth. This mixed up solution was then placed on shaking machine at 37 °C for 18 h. After shaking, the process was further repeated upto four times for serial dilution by mixing 1 mL of solution from the flask and 9 mL of 0.3 mM PBS. Finally 1 mL of the solution of bacterial growth with different concentration was taken and placed onto an agar plate. After 24 h of incubation at 37.8 °C, the number of bacterial colonies formed on the agar plates were counted visually. Moreover, the antibacterial performance was established from obtained results and percentage bacterial reduction was calculated according to following equation.(1)$$R=\frac{\left(B-A\right)}{B}\times 100\%$$
where *R* is the percentage bacterial reduction, *B* and *A* are the number of residual colonies of before and after treated with ultrafine CA fibers.

### 2.5. Material Analysis

The surface morphology and average diameter of template sericin-AuNCs were analyzed using transmission electron microscope (TEM, H-800 Hitachi Ltd., Tokyo, Japan) operating at 200 kV. The average particle size of sericin Ag NCs was measured from individual particles on the TEM images. Surface morphology of ultrafine electrospun CA fibers and sericin-AgNCs incorporated into ultra-fine CA fibers was measured using Field emission scanning electronic microscopy (FE-SEM, S-4800 Hitachi Ltd., Tokyo, Japan). The average fiber diameter was measured using image J software and graphical presented by counting 50 distinct fibers from FE-SEM images. The SEM samples were sputtered with gold before examination. Chemical composition of sericin-AgNCs incorporated into ultrafine CA fiber was obtained using energy-dispersive X-ray spectroscopy (EDS, S-3000 N Hitachi Ltd., Tokyo, Japan) elemental analysis, showing carbon, oxygen and silver as the main elements. Fourier transform infrared spectroscopy (FTIR) measurement was analyzed with Thermo Fisher Scientific Inc., Waltham, MA, USA. All the FTIR samples 1–2 mg were ground with 0.1 g KBr and pressed into a pellet before testing. Finally, the crystallinity of ultra-fine CA fibers and sericin-AgNCs incorporated into ultra-fine CA fibers were analyzed using X-ray diffraction (XRD) model D/max-IIB, Rigaku. Samples were analyzed in the range of 10–50° at the scanning speed of 2θ = 4°/min. The ultra violet adsorption and fluorescence studies were evaluated using UV-visible spectrophotometer (Shimadzu UV2450, Kyoto, Japan) and UV-visible fluorescence spectrofluorometer (FLS920, Edinburgh, Livingston, UK) respectively.

## 3. Results and Discussion

### 3.1. Synthesis of Sericin-AgNCs 

The synthesis of sericin-AgNCs was accomplished through the one pot, green synthesis route. As illustrated in Figure 1, sericin is actually a functional protein created by *Bombyx mori* (silkworms) in the production of silk, which make up the layers found on top of the fibrin. There are three different types of sericin, including sericin A, sericin B and sericin C. Due to the disulfide bonds and free cysteine, sericin play an important role in directing the synthesis of AgNCs. In the synthetic process, Ag ions were first introduced into the solution of sericin, where upon some Ag ions interact with the thiol groups of sericin to form functional Ag thiolate intermediates. After reduction, the spatial conformation of protein is further destructed from alkaline condition, followed by the exposure of more thiol site. The phenolic group of Y converts into a negative phenolic ion that can reduce Ag ions to Ag atoms. Finally, the resultant Ag atoms aggregate to form Ag clusters stabilized by the NH groups of the sericin [19]. 

### 3.2. Physico-Chemical Analysis 

The Figure 2A,B shows that sericin-AgNCs had maximum absorbance value at 440 nm and had fluorescence emission peaks at 500 nm respectively. Hence, the lyophilized sericin-AgNCs powder maintained the corresponding fluorescence emission. It is worth mentioned that samples were stable in their aqueous solutions and powder at 4 °C in the dark room for more than 3 months.

The TEM technique was performed to assess the surface morphology and measure the size distribution of the sericin-AgNCs. Figure 3A shows the TEM image of sericin-AgNCs in which sericin-AgNCs got quasi-spherical and monodisperse having ultra-small size. Whereas the diameter distribution (Figure 3B) indicates that sericin-AgNCs have an average size of 2.5 ± 0.5 nm and obtained diameter distribution is in agreement with Gaussian fitting [9]. Moreover, the SEM images (Figure 3C) of ultrafine CA fibers revealed smooth surface fibrous morphology having average fiber diameter of 300 nm. After incorporation of sericin-AgNCs into ultrafine CA fibers, the surface morphology (Figure 3E) retains its smooth surface fibrous morphology with similar average fiber diameter of 300 nm due to very less amount of sericin-AgNCs incorporated into CA fibers [33]. 

The EDS was performed to confirm the presence of silver clusters in the ultrafine CA fibers after incorporating the sericin-AgNCs. Figure 4 shows the EDS elemental mapping and EDS spectrum of sericin-AgNCs incorporated ultrafine CA fibers. The EDS spectrum confirms the presence of silver clusters with 0.26% total weight of other components present in CA fibers. While the element mapping shows the uniform presence of silver clusters on the surface of CA fibers [34].

Figure 5 indicates XRD spectrum of electrospun CA fibers and incorporated sericin-AgNCs with CA fibers. The XRD structure of electrospun CA fibers membrane showed a wide peak at 23.2°, which explains an amorphous structure. While incorporating sericin-AgNCs, the diffractions peak shifted to about 20.6°, the comparative peak intensity showed a slightly decreases [35,36]. These results ensured the successful incorporation of sericin-AgNCs with in the electrospun CA fibers and explained that the AgNCs were actively interacting with the CA microstructure acting as nuclei sites during the spinning process to induce CA fibers crystallization.

FTIR spectrum was used to analyze the chemical variations of electrospun CA fibers after incorporated sericin-AgNCs and confirms the successful hydrolysis of sericin-AgNCs with in the CA fibers. Figure 6 shows the broad spectrum of pure CA fibers and CA fibers with sericin-AgNCs. The pure CA fibers showed main typical peaks at 1745 cm^−1^ (C=O), 1375 cm^−1^ (C-CH_3_) and 1227 cm^−1^ (C-O-C); corresponding to the vibrations of the acetate group of the CA fibers [37]. Furthermore, sericin-AgNCs incorporated into CA fibers showed a wide peak generated at 3345 cm^−1^ which corresponds the NH and OH groups of sericin [38], while the hydroxyl vibration at 3680 cm^−1^ was slightly decreased. In addition, a weak adsorption was found at 875 cm^−1^, which corresponded to the silver compound and showed the successful incorporation of sericin-AgNCs into the CA fibers [36]. 

### 3.3. Antibacterial Activity Assessment

The potential application of antibacterial activity on ultrafine CA fibers incorporated with sericin-AgNCs was examined against *E. coli* and *S. aureus*. As prepared electrospun CA fibers without sericin-AgNCs were examined as a control experiment. The bacterial colonies were incubated in a growing medium with the presence of various amount of sericin-AgNCs incorporated into ultrafine CA fibers and the antibacterial characteristics of resultant products were examined by standard shake flask protocol. Increasing the amount of sericin-AgNCs into CA fibers showed significant improvement in antibacterial activity. The total density of bacterial colonies for pure CA fibers was very high (Figure 7A,B), and antibacterial activity was observed nearly zero, while the small amount of sericin-AgNCs incorporated into CA fibers showed a remarkable improvement against *E. coli* and *S. aureus*. Although the amount of sericin-AgNCs was only 0.17 mg/mL ultrafine CA fibers, the antibacterial rate against *E. coli* and *S. aureus* could reach more than 85% antibacterial reduction (Figure 8). Furthermore, the antibacterial effect of sericin-AgNCs mediated ultrafine CA fibers was significantly improved with increase amount of sericin-AgNCs. The results indicated that the 1.7 mg/mL amount of sericin-AgNCs was enough to get more than 99.9% antibacterial reduction rate (Figure 7A,B). Moreover, the further increased amount of sericin-AgNCs showed similar antibacterial reduction rate with 1.7 mg/mL sericin-AgNCs and results were calculated in Figure 8. These results suggest that pure CA fiber did not show any antibacterial reduction and it was believed that due to addition of sericin-AgNCs into ultrafine CA fibers was the reason behind of this remarkable antibacterial reduction. Additionally, literature confirms that the antibacterial reduction of any product containing Ag was mainly verified by the total release rate of Ag ions [19]. In present work, due to ultrasmall particle size, the increase of Ag amount will lead a greater number of silver ions to enlargement and increase the total surface area of Ag nanoparticle; hence the release rate of silver ions was enhanced which can ultimately enhance the antibacterial reduction rate using sericin-AgNCs/CA composite. As per our experimental outcomes, it can be summarized that the CA fibers containing 1.7 mg/mL sericin-AgNCs showed excellent antibacterial reduction rate and amount of Ag was much lower than used in previous Ag/composites reports [5,8,36,38,39]. While promising characteristics of sericin makes our product very potential against numerous applications including biomedical materials, sports apparatus, and laminating films. 

It is worth noting that silver nanoclusters compared to other metals were considered safer antimicrobial agents due to their low toxicity. Sericin-AgNCs showed remarkable antibacterial property with ultra low amount of 1.7 mg/mL and results suggested that sericin-AgNCs incorporated into ultrafine CA fibers can be potential candidate for many biomedical applications including wound healing. To address potential impact of sericin-AgNCs, more and more research should be carried out to highlight its side effects on human health and ecology. This will lead to a better understanding and facilitate use of silver nanoclusters on commercial large scale as antimicrobial agents for applications onto different biomedical applications. 

## 4. Conclusions

In the present study, a simple green synthesis route for sericin-AgNCs has been introduced and successfully incorporated into ultrafine CA fibers via electrospinning technique in order to report antibacterial properties. The average particle size of sericin-AgNCs was measured 2.5 nm and EDS results exhibited successful incorporation of sericin-AgNCs into ultrafine CA fibers. FTIR results confirmed the presence of sericin-AgNCs incorporated within ultrafine CA fibers. The antibacterial activities of sericin-AgNCs mediated ultrafine CA fibers were examined by shake flask method against the *E. coli* and *S. aureus* protocols. The results showed that sericin-AgNCs loaded ultrafine CA fibers exhibits outstanding antibacterial properties even with a ultralow content of sericin-AgNCs; as inhibition rate attained more than 90% at 0.17 mg/mL loading of sericin-AgNCs into ultrafine CA fibers and was finally reached >99.9% at 1.7 mg/mL amount of sericin-AgNCs against *E. coli* and *S. aureus*. The current results confirmed that the synergistic antibacterial effects of sericin-AgNCs with ultrafine CA fibers, and the composite can be used as promising antibacterial material for further study in biomedical applications. 

## Figures and Tables

**Figure 1 polymers-13-01411-f001:**
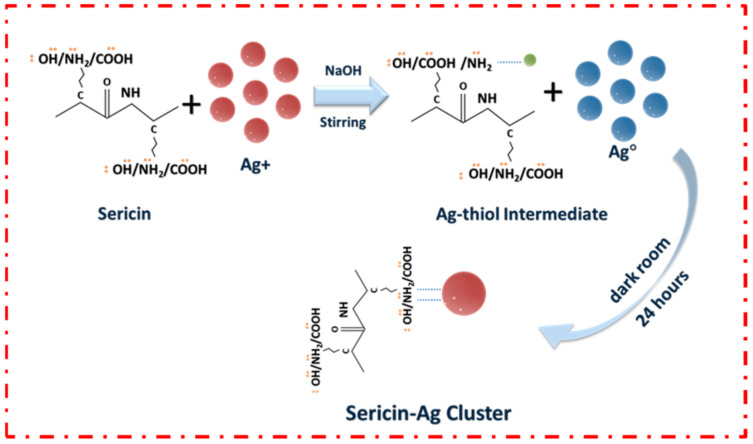
Typical illustration of synthesized sericin-AgNCs.

**Figure 2 polymers-13-01411-f002:**
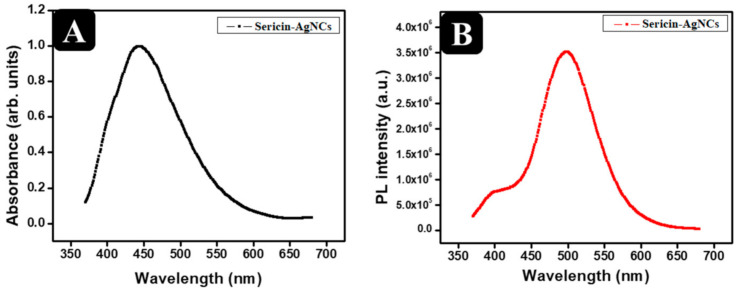
(**A**) UV-visible spectrum of sericin-AgNCs solution, and (**B**) UV-fluorescence spectrum of sericin-AgNCs solution.

**Figure 3 polymers-13-01411-f003:**
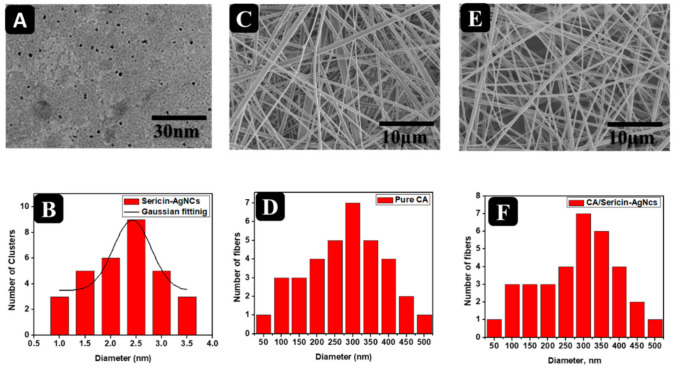
(**A**) TEM image of sericin-AgNCs, (**B**) histogram of sericin-AgNCs, (**C**) SEM images of pure CA fibers, (**D**) histogram of pure CA fibers, (**E**) SEM image of sericin-AgNCs incorporated into CA fibers, and (**F**) histogram of sericin-AgNCs incorporated into CA fibers.

**Figure 4 polymers-13-01411-f004:**
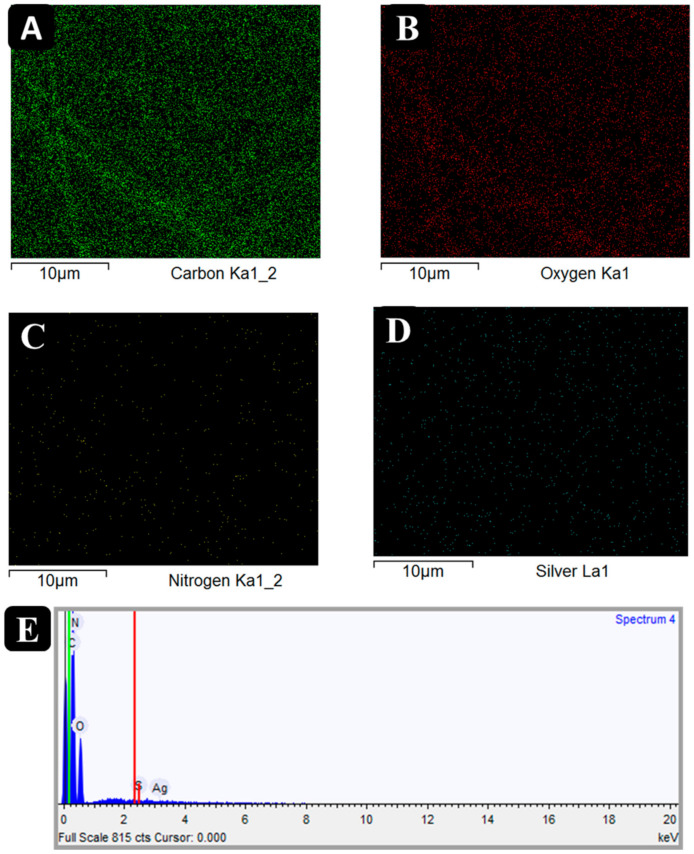
(**A**–**D**) EDS elemental mapping of sericin-AgNCs incorporated into ultrafine CA fibers, and (**E**) EDS spectrum and table (inset) showing the elemental composition of sericin-AgNCs incorporated into ultrafine CA fibers.

**Figure 5 polymers-13-01411-f005:**
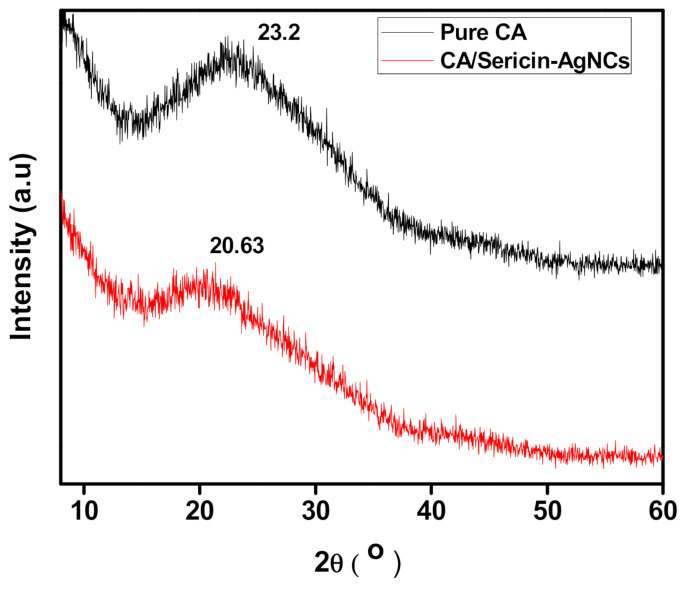
XRD results of pure CA and CA/sericin-AgNCs composite.

**Figure 6 polymers-13-01411-f006:**
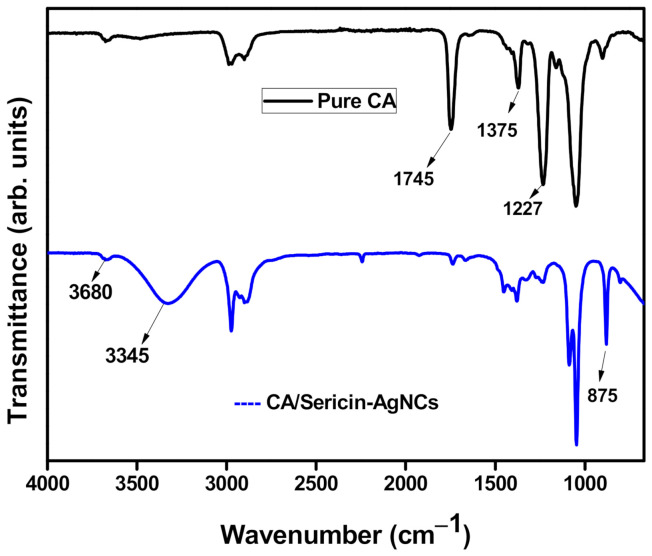
FTIR results of pure CA and sericin-AgNCs/CA fibers and both curves are shifted vertically for clarity.

**Figure 7 polymers-13-01411-f007:**
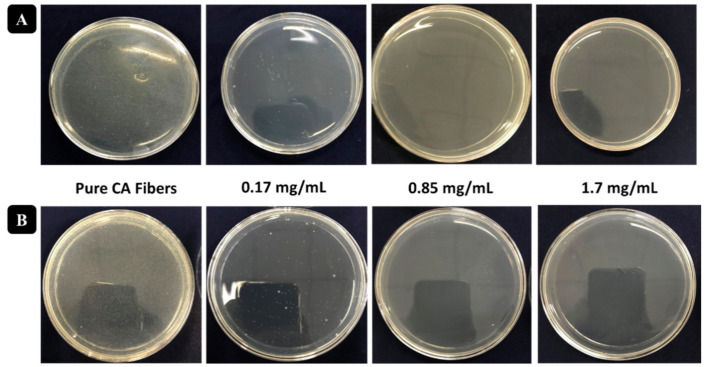
Bacterial colonies in the plates for (**A**) *Escherichia coli* and (**B**) *Staphylococcus aureus* at different contents of the sericin-AgNCs loaded into ultrafine CA fibers in the nutrient broth.

**Figure 8 polymers-13-01411-f008:**
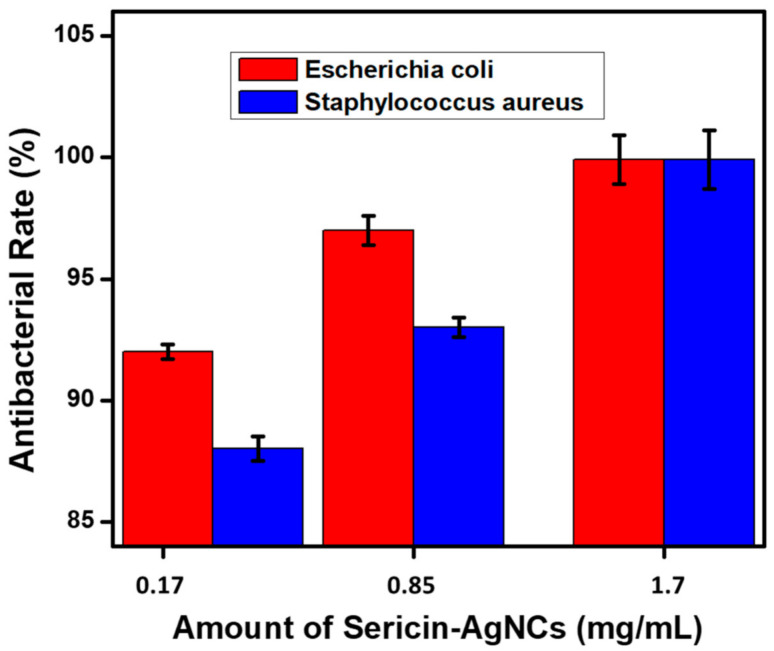
Degree of growth inhibition of different contents of sericin-AgNCs for bacterial suspension against *Escherichia coli* and *Staphylococcus aureus*.

## Data Availability

The data presented in this study are available on request from the corresponding author.

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
