# Peer review of "Green Synthesis and Incorporation of Sericin Silver Nanoclusters into Electrospun Ultrafine Cellulose Acetate Fibers for Anti-Bacterial Applications"

_polymers, 2021, doi:10.3390/polym13091411_

Round 1

Reviewer 1 Report

In this study, the authors prepared S-Ag NCs incorporated into CA fibers for potential antibacterial activity. In general, the article can be reconsidered for publication after a major revision. 

1) The manuscript should be polished by a native speaker. Indicate the purity of all consumables, it is important for reproducibility purposes. 
2) What kind of E.coli and S.aureus strains were used? There are many of them. 
3) Provide high-resolution SEM images. The authors stated that EDS detected Ag ions? It should be Ag NCs, right? Supply selected area EDS mapping to confirm the uniform presence of Ag in fibers. 
4) What was the final concentration of Ag in prepared fibers. Use some quantitative methods for analysis. 
5) Indicate the concentrations of S-Ag NCs shown in Fig. 5 & 6
6) Statistical analysis was not performed, each sample should be tested at least 3 times, and mean value + SD should be reported. 
7) What was the MIC of S-Ag NCs in respect to Ag concentration (for this, question 5 should be answered firstly). 
8) Some highly relevant recent studies should be added, these works are highly recommended: 
-Cetyltrimethylammonium Bromide (CTAB)-Loaded SiO2–Ag Mesoporous Nanocomposite as an Efficient Antibacterial Agent, Nanomaterials 2021, 11, 477 
- A Facile Method for the Fabrication of Silver Nanoparticles Surface Decorated Polyvinyl Alcohol Electrospun Nanofibers and Controllable Antibacterial Activities, Polymers 2020, 12, 2486

Author Response

In this study, the authors prepared S-Ag NCs incorporated into CA fibers for potential antibacterial activity. In general, the article can be reconsidered for publication after a major revision. 

Response: Authors are very thankful to reviewer for very positive comments, highlighted important scientific suggestions and considering our work for possible publication. We have tried our best to revise our manuscript according to reviewer suggestions. Following, we have discussed each comment in detail.

1) The manuscript should be polished by a native speaker. Indicate the purity of all consumables, it is important for reproducibility purposes.

 Answer: We thanks to the reviewer for suggestion. We have well revised our manuscript and counter check from a native English Speaker for improve writing skills. Also, we have provided the details purity of all consumables in our revised manuscript.

2) What kind of E.coli and S.aureus strains were used? There are many of them. 

Answer: We thanks to the reviewer for highlight important suggestion. We have used Escherichia coli (BUU25113) and Staphylococcus aureus (B-sub 168) for this experiment. Further, we have mentioned in our revised manuscript.

3) Provide high-resolution SEM images. The authors stated that EDS detected Ag ions? It should be Ag NCs, right? Supply selected area EDS mapping to confirm the uniform presence of Ag in fibers.

Answer: We thanks to the reviewer for highlight important suggestion. We have revised the SEM images and provided the high-resolution SEM images. While we have corrected the ‘Ag ions’ with Ag NCs. Also, we have provided EDS mapping to confirm the uniform presence of Ag in fibers.

 4) What was the final concentration of Ag in prepared fibers. Use some quantitative methods for analysis.

 Answer: We thanks to the reviewer for highlight important suggestion. We have used maximum concentration of 1.7 mg/mL sericin-AgNCs and well mentioned in our revised manuscript. Also, we have discussed quantitative analysis in Figure 5 and 6.

5) Indicate the concentrations of S-Ag NCs shown in Fig. 5 & 6

Answer: We thanks to the reviewer for highlight important suggestion. We have indicated the concentrations of sericin-AgNCs shown in revised figure 5.

6) Statistical analysis was not performed, each sample should be tested at least 3 times, and mean value + SD should be reported. 

Answer: We thanks to the reviewer for highlight important suggestion. We have performed each experiment for 3 times and mean value with error bars has been provided in revised manuscript.

7) What was the MIC of S-Ag NCs in respect to Ag concentration (for this, question 5 should be answered firstly).

We thanks to reviewer for kind suggestion. We have provided the MIC and MBC values as 4×103cfu/mL and 1.7 mg/mL respectively in our revised manuscript.

8) Some highly relevant recent studies should be added, these works are highly recommended: 
-Cetyltrimethylammonium Bromide (CTAB)-Loaded SiO2–Ag Mesoporous Nanocomposite as an Efficient Antibacterial Agent, Nanomaterials 2021, 11, 477 
- A Facile Method for the Fabrication of Silver Nanoparticles Surface Decorated Polyvinyl Alcohol Electrospun Nanofibers and Controllable Antibacterial Activities, Polymers 2020, 12, 2486

Answer: We thanks to the reviewer for kind suggestions. We have cited the suggested articles in our revised manuscript.

Reviewer 2 Report

Incorporation of the antibacterial complex, namely “Sericin encapsulated Silver Nanoclusters” (here Ser-AgNC), into ultrathin CA fibers, is the novel and simultaneously topical approach of the Mehdy et al to the development of antibacterial ultrafine polymer system. The essential merit of the submission is the nanotechnological methodology presented by electrospinning.

To reach the reasonable findings in antibacterial behavior of modified CA nanofibers, the nonspecific physical (TEM, field emission SEM, FTIR, WXRD, and XPS) and biologically specific shake flask method focused on antibacterial activity of the fibers have been used. In the latter case, for E.coli and S.aureus, the synergistic effect of two bactericidal components, sericin and AgNC, embedded in CA nanoarrays, has been clearly shown. This finding allows the authors to make the conclusion – after additional biological assessment, the fibers may be used as antibacterial material, in perspective.

The content of this paper falls within the scope of the Polymers spanning the biomedical polymers partition. With appropriate terminology and reasonable argumentation, the manuscript shows clearly the modification of the initial polymer surface/volume and followed changing the CA structure for example promoting the cellulose derivative crystallization. The manuscript abstract reflects the general issues of the paper. The literature cited is quite relevant to this study and the illustrations (the tables and the figures) are executed in an unambiguous and accurate manner with a reasonable interpretation.

Simultaneously, a number of flaws and typos have been met in the text.

Abbreviation “Sericin-AgNC” introduced in Abstract does not correspond to the abbreviation presented just below (L14) - “S-AgNC”

The introduction should essentially be expanded to present the behavior of sericin as the therapeutic and antibacterial polymer and especially to describe the composites of the silk protein with the other polymers such as fibroin/sericin, chitosan/sericin, etc. More important for this manuscript to assess the electrospun composites such as electrospun sericin/PLA fibers [Chao et al. 2018,  https://doi.org/10.1016/j.jddst.2018.01.022] [Zhao et al. 2014, http://dx.doi.org/10.1016/j.ijbiomac.2014.04.029], etc.

In the legend of Scheme 1, the authors should describe the red circle as the Ag-cluster or something else.

LL 238, 265: “sercin” should be considered as a typo.

- The same for Fig. 4, the letter “i” is omitted

- Fig. 6. The authors should disclose the letter-designations

L31: decants = decades(?)

L42: “reviled”; I guess it is the typo but it sounds quite dramatic, please replace  “revealed”.

L125: elecmental = elemental (?)

L125 and following EDS, please provide the meaning

L141: The phrase “After the addition of the addition of NaOH” should be edited.

L141: “to form bidentate Ag thiolate intermediates”. It looks like slang, please use another term e.g. bifunctional.

In conclusion, the authors should carefully check the whole text, enhance the idea of physical modification by the bactericide nano compound, and expand the introduction.

Major revision.

Author Response

Incorporation of the antibacterial complex, namely “Sericin encapsulated Silver Nanoclusters” (here Ser-AgNC), into ultrathin CA fibers, is the novel and simultaneously topical approach of the Mehdy et al to the development of antibacterial ultrafine polymer system. The essential merit of the submission is the nanotechnological methodology presented by electrospinning.

To reach the reasonable findings in antibacterial behavior of modified CA nanofibers, the nonspecific physical (TEM, field emission SEM, FTIR, WXRD, and XPS) and biologically specific shake flask method focused on antibacterial activity of the fibers have been used. In the latter case, for E.coli and S.aureus, the synergistic effect of two bactericidal components, sericin and AgNC, embedded in CA nanoarrays, has been clearly shown. This finding allows the authors to make the conclusion – after additional biological assessment, the fibers may be used as antibacterial material, in perspective.

The content of this paper falls within the scope of the Polymers spanning the biomedical polymers partition. With appropriate terminology and reasonable argumentation, the manuscript shows clearly the modification of the initial polymer surface/volume and followed changing the CA structure for example promoting the cellulose derivative crystallization. The manuscript abstract reflects the general issues of the paper. The literature cited is quite relevant to this study and the illustrations (the tables and the figures) are executed in an unambiguous and accurate manner with a reasonable interpretation.

Simultaneously, a number of flaws and typos have been met in the text.

Response: Authors are very thankful to reviewer for very positive comments, detailed discussion, and highlighted important scientific suggestions. We have tried our best to revise our manuscript according to reviewer suggestions. Following, we have discussed each comment in detail.

1) Abbreviation “Sericin-AgNC” introduced in Abstract does not correspond to the abbreviation presented just below (L14) - “S-AgNC”

Answer: We thanks to the reviewer for highlight important suggestion. We have corrected the abbreviation “S-AgNCs” with “sericin-AgNCs” in the abstract and counter check entire manuscript.

2) The introduction should essentially be expanded to present the behavior of sericin as the therapeutic and antibacterial polymer and especially to describe the composites of the silk protein with the other polymers such as fibroin/sericin, chitosan/sericin, etc. More important for this manuscript to assess the electrospun composites such as electrospun sericin/PLA fibers [Chao et al. 2018,  https://doi.org/10.1016/j.jddst.2018.01.022] [Zhao et al. 2014, http://dx.doi.org/10.1016/j.ijbiomac.2014.04.029], etc.

Answer: We thanks to the reviewer for kind suggestions. We have well revised the introduction part accordingly and expanded to discuss the importance of sericin material. Also, we have cited the suggested articles in our revised manuscript.

3) In the legend of Scheme 1, the authors should describe the red circle as the Ag-cluster or something else.

Answer: We thanks to the reviewer for kind suggestions. We have well revised the scheme 1 in our revised manuscript and Ag-cluster has been corrected.

4) LL 238, 265: “sercin” should be considered as a typo.

Answer: We thanks to the reviewer for highlight important typo mistake. We have corrected the word “sercin” with “sericin” in entire manuscript. 

5) - The same for Fig. 4, the letter “i” is omitted

Answer: We thanks to the reviewer for highlight important typo mistake. We have corrected the figure 4 accordingly. 

6)  Fig. 6. The authors should disclose the letter-designations

Answer: We thanks to the reviewer for highlight important suggestion. We have indicated the concentrations of sericin-AgNCs shown in revised figure 5 and improve caption writing.

7) L31: decants = decades(?)

Answer: We thanks to the reviewer for highlight important typo mistake. We have corrected the word “decants” with “decades” in our revised manuscript. 

8) L42: “reviled”; I guess it is the typo but it sounds quite dramatic, please replace  “revealed”.

Answer: We thanks to the reviewer for highlight important typo mistake. We have corrected the word “reviled” with “revealed” in our revised manuscript. 

9) L125: elecmental = elemental (?)

Answer: We thanks to the reviewer for highlight important typo mistake. We have corrected the word “elecmental” with “elemental” in our revised manuscript. 

10) L125 and following EDS, please provide the meaning

Answer: We thanks to the reviewer for highlight important suggestion. We have provided full abbreviation of EDS (Energy-dispersive X-ray spectroscopy) in our revised manuscript.

11) L141: The phrase “After the addition of the addition of NaOH” should be edited.

Answer: We thanks to the reviewer for highlight important suggestion. We have well corrected the sentence.

12) L141: “to form bidentate Ag thiolate intermediates”. It looks like slang, please use another term e.g. bifunctional.

Answer: We thanks to the reviewer for highlight important suggestion. We have well corrected the suggested sentence accordingly.

13) In conclusion, the authors should carefully check the whole text, enhance the idea of physical modification by the bactericide nano compound, and expand the introduction.

Answer: We thanks to reviewer for kind suggestion. We have improved the entire manuscript and try to enhance the idea, importance and expand the introduction with relevant literature.

Reviewer 3 Report

The manuscript entitled Green Synthesis of Sericin Silver Nanoclusters Incorporated into Electrospun Ultrafine CA Fibers for Anti-bacterial Applications submitted to Polymers Journal.

The concept of the manuscript is novel, fits and suitable to publish in Polymers Journal. This manuscript is generally well written and clearly presented however still needs to address many comments, major attention should be on English and grammar as well as scientific writing  and thus require substantial major revision.

  • English and grammar mistakes are present. The author should check the manuscript by native English Speaker to improve the quality of the manuscript.
  • Title need to be modified which can describe the whole research work. Don’t use any abbreviation in the title.
  • Provide a nice graphical abstract representing the overview of the MS with key highlights.
  • Keywords should be rewrite for example how antibacterial can be a keyword? it should be antibacterial activity or application.
  • In the introduction section, write the novelty of the work and the problem statement clearly. The manuscript is lacking to cite essential review of literature author should refer an cite some recent research articles Line no 32 refer and cite Journal of environmental management 223, 1086-1097, 2018; line no 37 for detailed mode of silver NPs mechanism refer and cite important article Environmental Science and Pollution Research 25 (11), 10250-10263.
  • More discussion on green synthesis using different biomolecules for example lignin, grape pomace phytochemicals add more details by referning  International journal of biological macromolecules 128, 391-400, 2019; Nanomaterials 10 (8), 1457, 2020. The detailed discussion about the novelty, significance of your research work and research gap relative to the literature is essential.
  • Alongwith the mentioned polymers recently some investigators used PHA-nanocomposites for antibacterial as well as other biomedical applications refer and cite the recent review article Bioresource technology, 124685, 2021. Microorganism name should be in italic form
  • Section 2.1 gives details of manufacturers for chemicals, instruments etc. All units should be checked and it should be in standard format.,
  • Section 2.,5 Give details of analysis . Statistical analysis of the results should be provided in the materials and methods section. It's important for all experimental work Report these values in the results and discussion.
  • TEM image is not visible to provide a new one, how do they determine particle size.
  • Have authors checked the stability of synthesized nanoclusters give details
  • Authors should determine MIC and MBC values of synthesized nanocomposites against the selected strain
  • Give values to the peak of FTIR. Substantial discussion of peaks and their comparison with the literature is expected during revision.
  • For figure and table captions give all details which is quite expected. Figure 7 give full name of microorganism used
  • Write the practical applications and future research perspectives and challenges by adding a new section before conclusions
  • The conclusion of the study is not discussed with the specific output obtained from the study, it could be modified with precise outcomes with a take home message. 

Author Response

The concept of the manuscript is novel, fits and suitable to publish in Polymers Journal. This manuscript is generally well written and clearly presented however still needs to address many comments, major attention should be on English and grammar as well as scientific writing and thus require substantial major revision.

Response: Authors are very thankful to reviewer for very positive comments and highlighted important scientific suggestions. We have tried our best to revise our manuscript according to reviewer suggestions. Following, we have discussed each comment in detail.

Comment # 1: English and grammar mistakes are present. The author should check the manuscript by native English Speaker to improve the quality of the manuscript.

Answer: We thanks to the reviewer for suggestion. We have well corrected entire English and grammar in our revised manuscript and counter check from a native English Speaker.

Comment # 2: Title need to be modified which can describe the whole research work. Don’t use any abbreviation in the title.

Answer: We thanks to the reviewer for highlight important suggestion. We have well revised the title in our manuscript.

Comment # 3: Provide a nice graphical abstract representing the overview of the MS with key highlights.

Answer: We thanks to the reviewer for kind suggestion. We have drawn a graphical illustration of our work.

Comment # 4: Keywords should be rewrite for example how antibacterial can be a keyword? it should be antibacterial activity or application.

Answer: We thanks to the reviewer for highlight important correction. We have well revised the keywords in our revised manuscript.

Comment # 5: In the introduction section, write the novelty of the work and the problem statement clearly. The manuscript is lacking to cite essential review of literature author should refer an cite some recent research articles Line no 32 refer and cite Journal of environmental management 223, 1086-1097, 2018; line no 37 for detailed mode of silver NPs mechanism refer and cite important article Environmental Science and Pollution Research 25 (11), 10250-10263.

Answer: We thanks to the reviewer for kind suggestions. We have well revised the introduction part accordingly and cited the suggested articles in our revised manuscript.

Comments # 6: More discussion on green synthesis using different biomolecules for example lignin, grape pomace phytochemicals add more details by referring International journal of biological macromolecules 128, 391-400, 2019; Nanomaterials 10 (8), 1457, 2020. The detailed discussion about the novelty, significance of your research work and research gap relative to the literature is essential.

Answer: We thanks to the reviewer for kind suggestions. We have discussed further on green synthesis and cited the suggested articles in our revised manuscript.

Comment # 7: Along with the mentioned polymers recently some investigators used PHA-nanocomposites for antibacterial as well as other biomedical applications refer and cite the recent review article Bioresource technology, 124685, 2021. Microorganism name should be in italic form

Answer: We thanks to the reviewer for kind suggestions. We have cited the suggested articles in our revised manuscript and corrected the microorganism names in italic form.

Comment # 8: Section 2.1 gives details of manufacturers for chemicals, instruments etc. All units should be checked, and it should be in standard format.

Answer: We thanks to the reviewer for kind suggestions. We have provided details of each manufacturer for chemicals and instruments. Also, counter checked all the units and corrected in standard format.

Comment # 9: Section 2.,5 Give details of analysis. Statistical analysis of the results should be provided in the materials and methods section. It's important for all experimental work Report these values in the results and discussion.

Answer: We thanks to the reviewer for kind suggestions. We have provided details of analysis in revised manuscript and clearly discussed.

Comment # 10: TEM image is not visible to provide a new one, how do they determine particle size.

Answer: We thanks to the reviewer for kind suggestion. We have taken again a clear TEM image of sericin AgNCs and average particle size was measured from TEM image using image J software.

Comment # 11 Have authors checked the stability of synthesized nanoclusters give details

Answer: We thanks to reviewer for kind suggestion. Although our synthesized nanoclusters are stable, but we cannot be able to provide details this time and we will take this suggestion for future projects.

Comment # 12: Authors should determine MIC and MBC values of synthesized nanocomposites against the selected strain

Answer: We thanks to reviewer for kind suggestion. We have provided the MIC and MBC values as 4×103cfu/mL and 1.7 mg/mL respectively in our revised manuscript. For details study, we will take care and keep in mind for future projects.

Comment # 13: Give values to the peak of FTIR. Substantial discussion of peaks and their comparison with the literature is expected during revision.

Answer: We thank to reviewer for kind suggestion. We have well revise the FTIR graph, mention the peak values and discussed in revise manuscript.

Comment # 14: For figure and table captions give all details which is quite expected. Figure 7 give full name of microorganism used

Answer: We thanks to reviewer for kind suggestion. We have revised the captions and mentioned the full name of microorganism used in Figure 7.

Comment # 15: Write the practical applications and future research perspectives and challenges by adding a new section before conclusions

Answer: We thanks to reviewer for kind suggestion. We have discussed the possible application and future research perspectives before the conclusion.

Comment # 16: The conclusion of the study is not discussed with the specific output obtained from the study, it could be modified with precise outcomes with a take home message. 

Answer: We thanks to reviewer for kind suggestion. We have improved the conclusion accordingly in our revised manuscript.

Round 2

Reviewer 1 Report

The manuscript has been revised properly, hence, it can be considered for publication

Author Response

Authors are very thankful to reviewer for very positive comments and highlighted important scientific suggestions to improve our manuscript. 

Reviewer 2 Report

The content of the submission was improved, the legends were amended. The text is much more readable.

I advise going to further promotion of manuscript.  

Author Response

(The authors gave the same response as above.)

Reviewer 3 Report

Authors have substantially revised the manuscript according to the comments however still some essential information is required before its acceptance.

How authors determined the MIC and MBC the mentioned values in the response is for which strain? Give details for the sam.

why authors not provided the details of stability of synthesized nanoclusters. They are unable what it means?

some units are not in correct form and some spelling mistakes are there

Author Response

Authors have substantially revised the manuscript according to the comments however still some essential information is required before its acceptance.

Response: Authors are very thankful to reviewer for very positive comments and highlighted important scientific suggestions. We have tried our best to revise our manuscript according to reviewer suggestions. Following, we have discussed each comment in detail.

1) How authors determined the MIC and MBC the mentioned values in the response is for which strain? Give details for the sam.

Answer: We thanks to reviewer for kind suggestion. We determined the MIC values by following broth dilution method and number of bacterial colonies formed on the agar plates were counted visually followed by percentage bacterial reduction equation. And we assess the values against both strains Escherichia coli and Staphylococcus aureus. Each value is provided in Figure 8 and well discussed in section 3.3. 

2) why authors not provided the details of stability of synthesized nanoclusters. They are unable what it means?

Answer: We thanks to the reviewer for kind suggestion. We have provided the stability analysis of synthesized nanoclusters in revised manuscript. Please find the revised figure 2 and its discussion in 3.2 section.

3) some units are not in correct form and some spelling mistakes are there

Answer: We thanks to the reviewer for highlight typo mistakes. We have well corrected the units and spelling mistakes in our revised manuscript.
